# Surface Activity, Wetting, and Aggregation of a Perfluoropolyether Quaternary Ammonium Salt Surfactant with a Hydroxyethyl Group

**DOI:** 10.3390/molecules28207151

**Published:** 2023-10-18

**Authors:** Jiangxun Dou, Jiaoyan Liu, Yan Wang, Lifei Zhi, Jixian Shen, Guoyong Wang

**Affiliations:** 1School of Chemical Engineering and Technology, Taiyuan University of Science and Technology, Taiyuan 030024, China; djiangx@126.com (J.D.); ljyan0920@126.com (J.L.); lifeizhi@yeah.net (L.Z.); 2China Research Institute of Daily Chemical Industry, Taiyuan 030001, China; wang3063025@126.com

**Keywords:** fluorocarbon surfactant, quaternary ammonium salt, solution property

## Abstract

This paper reports the synthesis of a novel quaternary surfactant containing a hydroxyethyl group (PFPE-C) and the surface properties of its aqueous solution (investigated by comparisons with two structurally similar chemicals, dodecyl-(2-hydroxyethyl)-dimethylammonium chloride (DHDAC) and PFPE-A). The minimum surface tension ($\gamma_{CMC}$) and critical micelle concentration (CMC) of the PFPE-C aqueous solution were 17.35 mN/m and 0.024 mmol/L, respectively. This study confirms that surfactants containing hydroxyethyl groups efficiently reduce the surface tension of aqueous solutions, and fluorocarbon surfactants exhibit better surface activity than ordinary hydrocarbon surfactants with similar structures. The micellization, aggregation, air-water interfacial adsorption, and wettability of PFPE-C aqueous solutions have been systematically investigated. Highly concentrated PFPE-C aqueous solutions exhibit good wettability on PTFE and paraffin films. Moreover, the aggregates of PFPE-C in the aqueous solution were clearly seen as vesicles on Cryo-TEM micrographs. Primary biodegradation results indicate that 19% of PFPC-C can be degraded within one week.

## 1. Introduction

Surfactants are substances that are active at interfaces, have the ability to efficiently reduce interfacial tension and can change the interfacial composition and structure of a system [1]. In the past decades, surfactants with special structures and novel properties have gradually attracted attention, and fluoropolymers are a prominent candidate [2]. Fluorocarbon surfactants are special surfactants in which the introduced fluorine atom partially or completely replaces the hydrogen in the hydrophobic chain of the hydrocarbon surfactant to form a hydrophobic portion consisting of a fluoroalkyl or alkenyl group [3]. Unlike traditional hydrocarbon surfactants, fluorocarbon surfactants are widely used in firefighting [4], oil extraction [5,6], and electronics [7,8], and for manufacturing paper [9] and coatings [10,11], due to their excellent physicochemical properties, including high surface activity, thermal and chemical stability, and water and oil repellency [12,13].

Fluorocarbon surfactants are, by far, the most surface-active of all surfactants, with a mass concentration of 0.005–0.1% that would reduce the surface tension of water to below 20 mN/m [14], while hydrocarbon surfactants only reduce the surface tension of water to 30–40 mN/m because of the limitation of hydrophobic chain methylene groups. In addition, fluorocarbon surfactants could be adsorbed on the organic solvent-air interface as a single molecular membrane, reducing the surface tension of nonprotonic polar organic solvents such as aliphatic and aromatic, which is not a feature of hydrocarbon surfactants [15].

Fluorocarbon surfactants are thermally and chemically stable, due to the presence of fluorine atoms with strong electronegativity [16]. However, perfluorinated alkyl groups with carbon-fluorine chains containing seven or more carbon atoms undergo bioaccumulation; for example, Perfluorooctanoyl fluoride, sulfonic acid, sulfonates and related derivatives (collectively known as PFOS) are classified as persistent organic pollutants (POPs), and long-chain perfluoroalkyl sulfonic acids (PFAS) are difficult to degrade in the environment [17]. Although long-chain fluoropolymers pose serious environmental problems, their unparalleled properties are of great value in high-tech applications. To overcome this limitation, many scholars have proposed numerous coping strategies, such as biodegradation [18], photodegradation [19], acoustic degradation [20], or seeking alternatives to PFOS, among which, the replacement of perfluorinated chain segments with no-perfluorinated chain segments is an advantageous method for the synthesis of non-bioaccumulative fluorocarbons (NBCs). In a previous study, we introduce oxygen atoms into perfluorinated chain segments to synthesize a perfluoropolyether surfactant with degradable sites, and the oxygen heteroatom insertion also makes the fluorinated surfactant molecule more elastic and has a lower Krafft point [21].

However, the high cost of perfluoropolyether surfactants limits further applications. Therefore, reducing the dosage of perfluoropolyether surfactants is imperative for industrial applications. Wang et al. proved that hydrocarbon surfactants containing hydroxyethyl groups can reduce its critical micelle concentration; based on this fact, we would like to design a perfluoro-surfactant-containing hydroxyethyl group in order to improve its surface activity [22]. Therefore, in this study, a perfluoropolyether-quaternary ammonium salt-containing hydroxyl group (PFPE-C) has been designed, synthesized, and characterized from perfluoropolyether acyl fluoride (PFPF), *N*,*N*-dimethyl-1,3-propanediamine, and chloroethanol. The synthetic route of PFPE-C is shown in Figure 1; its physicochemical properties, such as air-water interfacial adsorption, aggregation and wettability were investigated systematically.

## 2. Experimental Section

### 2.1. Materials

2,5-Bis(trifluoromethyl)-3,6-dioxaundecafluorononanoyl fluoride (PFPF) was supplied by Lisheng Reagent Biochemistry Technology Co., Ltd. (Shanghai, China), *N*,*N*′-dimethyl-1,3-propyldiamine was supplied by Aladdin (Bay City, MI, USA), triethylamine, acetone, and diethyl ether were supplied by Tianjin Shentai Chemical Reagent Co. (Tianjin, China), and 2-chloroethanol was purchased from Shandong West Asia Chemical Industry Co. (Linyi, China). Deionized water (18.2 MΩ) was used during experimentation.

### 2.2. Synthesis of PFPE-C

First, the intermediate (perfluoropolyetheramidopropyl dimethyl tertiary amine) was synthesized by amidation, using PFPF and *N*,*N*′-dimethyl-1,3-propyldiamine as the raw materials and triethylamine as the acid binding agent. Subsequently, the quaternary amine salt, fluorosurfactant PFPE-C, was synthesized using chloroethanol as the quaternary amination reagent. The ether and triethylamine used for synthesis were dried before use. The detailed procedure is provided below.

First, *N*,*N*′-dimethyl-1,3-propyldiamine (5.2 g) and triethylamine (5.0 g) in 50 mL of ether were added into a 250 mL three-necked flask, followed by the gradual dropwise addition of PFPF (19.5 g) over a time period of 1–2 h under ice-bath conditions (0 °C). After restoring the reaction mixture to room temperature, its temperature was increased to 45 °C under nitrogen, and refluxed for more than 5 h. Subsequently, rotary evaporation was used to remove any excess solvent and unreacted materials, and the as-prepared solid was washed with deionized water (3–5 times) and dried to obtain the intermediate in more than 85% yield.

After this, chloroethanol (2.6 g) in 50 mL of anhydrous ethanol was transferred into a 250 mL three-neck flask, and the synthesized intermediate (17.4 g) was added into this solution. This reaction mixture was then heated to 65 °C and refluxed for more than 5 h. Finally, rotary evaporation was used to remove the solvent and excess chloroethanol, and a yellow waxy solid, PFPE-C, was obtained after drying; the final yield of the PFPE-C surfactant was 90 percent.

### 2.3. Characterization of PFPE-C

The intermediate and PFPE-C were structurally characterized by Fourier transform infrared (FT-IR), nuclear magnetic resonance (NMR) spectroscopy and Electrospray Ionization Mass Spectrometry (EIS-MS) using a VERTEX-70 (Bruker, Mannheim, Germany) infrared spectrometer, INOVA-400 (Bruker, Mannheim, Germany) nuclear magnetic resonance instrument and Thermo Scientific QE plus (Thermo Fisher Scientific, Waltham, MA, USA), respectively.

### 2.4. Experimental Methods

Surface tension analysis: The equilibrium surface tension was measured by the De-Nuöy ring approach with a Krüss K100 tensiometer (Krüss Company, Hamburg, Germany) at 25.0 ± 0.1 °C. All solutions were prepared and stored for a single day before experimentation to ensure steadiness, uniformity, and balance. For each sample concentration, the reported ST value indicates the mean value of three measurements to minimize errors.

A BP100 tensiometer (Krüss Company, Hamburg, Germany) was used at 25.0 ± 0.1 °C to record dynamic surface tension (DST) data. The effective surface ages were within the range of 10 ms to 200 s, with an accuracy of ±0.01 mN/m.

Electrical conductivity measurements: EC data were measured with a Leici conductivity analyzer (model DDS-11A, Shanghai Leici-Chuangyi Instrument and Meter Company, Shanghai, China). While recording data, the temperature was systematically managed by a circulating water bath with an accuracy of ±0.1 °C.

Contact angle measurements: The wettability of the PFPE-C on the low-energy surface of specific substrates (polytetrafluoroethylene (PTFE) and paraffin films) was measured by a DSA 25 instrument (Krüss Company, Hamburg, Germany). The temperature was maintained at 25 °C with a relative accuracy of 0.04% using a thermostatic water bath (Haake Company, Vreden, Germany).

Dynamic light scattering analysis: The actual radius and dimension distribution of PFPE-C aggregates in aqueous solution were measured by a Zeta Plus Particle Size Analyzer (Brookhaven, Holtsville, NY, USA). The samples were allowed to remain at room temperature for more than 5 h before the experiment. The experimental polydispersity index (PDI) value was 0.165.

Negative-stained and Cryo Transmission electron microscopy: A JEM-1011 (Jeol Company, Tokyo, Japan) instrument was used to investigate the morphology of PFPE-C aggregates in aqueous solution at 100 kV. Prior to TEM analysis, the samples were stained with 2 wt% of phosphotungstic acid on a carbon-coated copper mesh.

Using a Talos F200C TEM (Thermo Fisher, Waltham, MA, USA), we froze PFPE-C aggregates in an aqueous solution to reduce the damage caused on the sample with electron beam to obtain a more realistic morphology and structure from which more detailed studies can be performed, and the samples were left to stand for two weeks prior to the experiment.

Primary Biodegradation Experiment: The primary biodegradation experiment was performed according to the China National Standard GB/T 15818-2006 [23].

## 3. Results and Discussion

### 3.1. Preparation and Characterization of PFPE-C

The perfluoropolyether hydroxyethyl quaternary ammonium salt, PFPE-C, was synthesized from PFPE, *N*,*N*-dimethyl-1,3-propanediamine, and chloroethanol, using a two-step amidation and quaternization process, as shown in Figure 1. The structure of the intermediate and product was characterized by FT-IR, ^1^H-NMR and ESI-MS, as shown in Figure 1, Figure 2 and Figure 3.

Figure 1 shows the FT-IR spectrum of PFPE-C. It indicates a broad absorption at 3400 cm^−1^ (corresponding to the N–H stretching vibration), with peaks at 1708 cm^−1^ (corresponding to the C=O stretching vibration), 2850–2962 cm^−1^ (corresponding to –CH_3_ and –CH_2_– stretching vibrations), 1460 cm^−1^ (corresponding to –CH_3_ and –CH_2_– bending vibrations), 1236 cm^−1^ (corresponding to the C–F stretching vibration), and 1150 cm^−1^ (corresponding to the C–O–C stretching vibration).

Figure 2 shows the ^1^H NMR spectra of the intermediate and PFPE-C. The strongest peak at 2.04 ppm could be attributed to CD_3_COCD_3_, while the peak positions of the different hydrogen atoms could be inferred from their chemical shifts.

The relative molecular mass of the expected product PFPE-C is 660.774, from Figure 3, the peak with *m*/*z* of 625.10188 is the (M-Cl)^+^ ion peak. The combination of FT-IR,^1^H-NMR and ESI-MS shows that PFPE-C has been successfully synthesized.

### 3.2. Surface Activity

Figure 4 shows the variation of the surface tension with the concentration of PFPE-C in an aqueous solution at 25 °C. The equilibrium surface tension method was used to investigate surface activity. At low concentrations (less than 0.024 mmol/L), the surface tension of PFPE-C decreased linearly on increasing the concentration of PFPE-C, possibly due to the orientation arrangement of PFPE-C. At a certain concentration (concentration over 0.024 mmol/L), the surface tension remained almost constant, with a saturation of the surface adsorption. The breakpoint in the γ versus C plots is assigned to the CMC, which is consistent with commercial surfactant behavior. This was used to calculate the critical micelle concentration (CMC; 0.024 mmol/L) and lowest surface tension ($\gamma_{CMC}$; 17.35 mN/m) of the system.

In order to study the properties of PFPE-C surfactant in a deeper way, the surface activity of PFPE-C was compared to those of the perfluoropolyether quaternary ammonium salt PFPE-A (with an identical hydrophobic carbon-fluorine chain and different hydrophilic part compared to PFPE-C) and hydroxyethyl quaternary ammonium salt DHDAC(dodecyl-(2-hydroxyethyl)-dimethylammon-ium chloride; with a different hydrophobic part and identical hydrophilic part compared to PFPE-C) reported in the literature [21,22]. The chemical structural formulae and characteristic parameters of PFPE-C, PFPE-A and DHDAC surfactants are shown in Figure 2 and listed in Table 1, respectively.

The Gibbs adsorption isotherm was used to calculate the saturation adsorption amount $\Gamma_{max}$, and the cross-sectional area $A_{min}$, for each adsorbed surface-active molecule, as follows:(1)$$\Gamma_{max}=-\frac{1}{2.303nRT}{\left(\frac{\partial \gamma }{\partial \mathrm{lg}C}\right)}_{T}$$
(2)$$A_{min}=\frac{1}{N_{A}\Gamma_{max}}$$
where R is the molar gas constant (8.314 J/mol/K), T is the absolute temperature, γ is the surface tension of the aqueous solution of PFPE-C, C is the concentration of PFEP-C, and $N_{A}$ is the Avogadro’s constant (6.022 × 10^23^ mol^−1^).

As listed in Table 1, although the lowest surface tension of the aqueous solution of PFPE-C was slightly higher than that of PFPE-A, its critical micelle concentration was significantly lower. This indicates that a hydroxyethyl group in the hydrophilic part of a surfactant efficiently reduces the surface tension of aqueous solutions of the surfactant. Moreover, the saturation adsorption of PFPE-C was higher than that of PFPE-A, while its molecular cross-sectional area was lower. This indicates that PFPE-C molecules are more tightly arranged at the gas–liquid interface than PFPE-A molecules, possibly due to facile hydrogen-bond formation by the hydroxyethyl groups in the hydrophilic head of the former. Notably, the electrostatic repulsion between these hydrophilic-head groups affected the $\Gamma_{max}$ and $A_{min}$.

The $\gamma_{CMC}$ and CMC of the PFPE-C aqueous solution were both significantly lower than those of the DHDAC aqueous solution. This indicates that the surface properties of perfluoropolyether surfactants are better than those of ordinary hydrocarbon surfactants with similar structures, and confirms that the fluorocarbon chains in the hydrophobic part of the synthesized surfactant exhibit a strong hydrophobic effect.

### 3.3. Thermodynamic Parameters of PFPE-C Systems

In order to investigate the thermodynamic properties of PFPE-C micellization, the conductivity curves of aqueous solutions of PFPE-C at different temperatures vs. concentration were determined by the conductivity method. As shown in Figure 5, the conductivity increased linearly with the concentration of PFPE-C; this trend underwent a reversal at the CMC. The slope of the straight line after the CMC was smaller than that of the straight line before it, indicating a reduction in the mobility of free ions in the solution after micelle formation.

Table 2 shows the thermodynamic parameters of PFPE-C micellization at different temperatures. In the following equations, $\Delta G_{m}^{\theta }$ is the standard Gibbs free energy change of micelle formation, while $\Delta H_{m}^{\theta }$ and $\Delta S_{m}^{\theta }$ are the enthalpy and entropy change of micellization, respectively [24]:(3)$$\Delta G_{m}^{\theta }=\left(1+\beta \right)RT\ln X_{\mathrm{CMC}}$$
(4)$$\Delta H_{m}^{\theta }=-\left(1+\beta \right)RT^{2}\frac{\mathrm{dln}X_{\mathrm{CMC}}}{dT}$$
(5)$$\Delta S_{m}^{\theta }=\frac{\left(\Delta H_{m}^{\theta }-\Delta G_{m}^{\theta }\right)}{T}$$
where $X_{\mathrm{CMC}}$ (=CMC/55.4) is the CMC value converted to the volume molar fraction, β (=1 − α) indicates the counter ion binding of micelles, and the micelle dissociation (α) is the ratio of slopes before and after the inflection point (at the CMC) in the conductivity graph [25]. The results are summarized in Table 2.

As shown in Table 2, the CMC values measured by the conductivity method were identical to those obtained by the equilibrium surface tension method (Table 1), indicating that PFPE-C exhibits no pre-micellization behavior in aqueous solutions. Additionally, the CMC values increased on increasing the temperature, possibly due to the elevated temperature that weakened the interaction of the hydrophobic-chains and destabilized the structure of surfactant molecules during aggregation in aqueous solution. Moreover, the $\Delta G_{m}^{\theta }$ and $\Delta H_{m}^{\theta }$ values are negative, indicating that the process of micelle formation of PFPE-C in aqueous solutions is a spontaneous exothermic process, in which dispersion forces play a major role [26]. Additionally, the absolute value of $\;\Delta H_{m}^{\theta }$ increased on increasing the temperature, indicating that the number of water molecules bound to the head of the surfactant decreased at elevated temperatures, reducing the heat of dehydration during micelle formation. The value of $\Delta G_{m}^{\theta }$ also decreased on increasing the temperature, indicating that the tendency of spontaneous micelle formation increased at higher temperatures. According to Table 2, entropy is the main driving force for spontaneous micelle formation, and the driving force for the micellization of surfactants in aqueous solution is mainly the interaction between the hydrophobic chains of the surfactant molecules, which when aggregated causes the release of the hydration layer around the hydrophobic chains, leading to an increase in the entropy of the solution system [27].

### 3.4. Adsorption and Diffusion Kinetics of PFPE-C

To investigate the adsorption–diffusion kinetics of PFPE-C at the air–water interface, the dynamic surface tension of PFPE-C at different concentrations was investigated by the maximum bubble pressure method. As shown in Figure 6, the surface tension gradually decreased, finally attaining a stable value, as the surface age increased; this indicates that the PFPE-C molecules underwent diffusion until the formation of an equilibrium state at the interface. Higher concentrations of PFPE-C showed a more rapid reduction of surface tension. Notably, the effect of concentration on the surfactant diffusion–adsorption of PFPE-C was in accordance with Fick’s law.

According to the adsorption model proposed by Eastoe et al., the Ward–Tordai equation can explain the mass transfer process of surfactant molecules between the bulk phase and interface at surfactant concentrations lower than the CMC. However, this equation contains a convolution integral that cannot be solved; therefore, the asymptotic solution derived by Miller et al. for the Word–Tordai adsorption equation was used in this study [28]. The equation is as follows:(6)$$\Gamma \left(t\right)=2C_{0}\sqrt{\frac{\mathrm{Dt}}{\pi }}-2\sqrt{\frac{D}{\pi }}\int_{0}^{\sqrt{t}}C_{s}d\left(\sqrt{t-\tau }\right)$$
where $\;\Gamma \left(t\right)$ is the interfacial adsorption at moment t, $C_{0}$ is the concentration of the bulk phase, $C_{s}$ is the concentration of the subsurface layer, D is the apparent diffusion coefficient, π is 3.142, τ is a dummy variable, $2C_{0}\sqrt{\frac{\mathrm{Dt}}{\pi }}$ indicates the migration of surfactant molecules from the bulk phase to the subsurface layer, and $2\sqrt{\frac{D}{\pi }}\int_{0}^{\sqrt{t}}C_{s}d\left(\sqrt{t-\tau }\right)$ represents the migration of surfactant molecules from the bulk phase to the subsurface layer as the concentration increases, with the reverse diffusion of surfactant molecules migrating from the subsurface layer back to the bulk phase.
(7)$$\text{Short-time}\;\mathrm{approximation},\;t\to 0\;\;\;\gamma {\left(t\right)}_{t\to 0}=\gamma_{0}-2nRTC_{0}\sqrt{\frac{\mathrm{Dt}}{\pi }}$$
(8)$$\text{Long-time}\;\mathrm{approximation},\;t\to \infty \;\;\;\gamma {\left(t\right)}_{t\to \infty }=\gamma_{\mathrm{eq}}+\frac{nRT\Gamma_{\mathrm{eq}}^{2}}{C_{0}}\sqrt{\frac{\pi }{4\mathrm{Dt}}}$$

As shown in Equations (7) and (8), the t→0 condition indicates a state without reverse diffusion; here, the surfactant solution is considered to be a dilute solution; at t→∞, the subsurface layer concentration $C_{s}$, is infinitely close to the bulk phase concentration [19].

Here, T is the absolute temperature, $\gamma_{0}$ is the surface tension of the pure solvent, $\gamma \left(t\right)$ is the dynamic surface tension at moment t, $\gamma_{\mathrm{eq}}$ is the equilibrium surface tension, and $\Gamma_{\mathrm{eq}}$ is the adsorption at the surface tension equilibrium; the value of n is considered to be 2 for ionic surfactants. Equations (7) and (8) indicate that $\gamma {\left(t\right)}_{t\to 0}$ is linearly related to t^1/2^, with an intercept of $\gamma_{0}$, while $\gamma {\left(t\right)}_{t\to \infty }$ is linearly related to t^−1/2^, with an intercept of $\gamma_{\mathrm{eq}}$. 

Figure 7 shows the dynamic surface tension curves of PFPE-C vs. t^1/2^ and t^−1/2^; the diffusion coefficients could be calculated from the slope of the curves, where D_eff_ and D_s_ represent the diffusion coefficients of the short-time and long-time adsorption mechanism, respectively. The results are shown in Table 3.

As shown in Figure 7, the slopes of the short-time (a) and long-time (b) adsorption mechanism curves decreased on increasing the surfactant concentration; this is consistent with the increasingly rapid surface-tension reduction at higher concentrations shown in Figure 6. Moreover, the diffusion coefficients D_eff_ and D_t_ decreased on increasing the surfactant concentration, indicating greater surfactant–molecule electrostatic repulsions at higher concentrations, which limits the free movement of molecules. Notably, the D_eff_/D_t_ ratio indicates a hybrid kinetic-controlled adsorption process [29].

### 3.5. Aggregation Behavior

To investigate the aggregation behavior of PFPE-C in aqueous solutions, its aggregated particle size and microstructure were investigated by DLS and TEM. Figure 8 shows the intensity-weighted size distribution (a) and negative staining TEM images (b) for PFPE-C aqueous solution at a concentration of 0.1 wt%, and Cryo-TEM images (c) at a concentration of 2.5 wt%.

Figure 8a indicates that the aggregate size showed a unimodal distribution at 440 nm, with diameters in the range of 425–460 nm. This is consistent with the TEM image (Figure 8b,c), which indicates that the PFPE-C surfactant forms vesicular aggregates with a hollow-sphere structure in aqueous solution. It is widely known that the aggregation behavior of surfactants in aqueous solution is mainly driven by hydrophobic forces; therefore fluorocarbon surfactants with bulky hydrophobic chains have excellent hydrophobicity, which makes them more likely to form complex structural aggregates [30,31]. However, aggregates with banded structures can be seen in the upper right corner of Figure 8c, and the reason for their creation has not yet been concluded.

### 3.6. Wettability of PFPE-C-Based Solutions on Low-Energy Surfaces

Dynamic contact angle measurements by the seated drop method using PTFE and paraffin membranes (as hydrophobic surfaces) were used to investigate the wettability of PFPE-C aqueous solutions. As shown in Figure 9, the contact angle of PFPE-C on PTFE and paraffin films gradually decreased with time. Notably, for concentrations lower than the CMC, the contact angle changed negligibly with time, possibly because the surfactant molecules were not closely aligned at the gas–liquid interface; this generated a large surface tension at the gas–liquid interface, forming a low-ductility adhesion layer.

In this study, at low concentrations of PFPE-C (less than 1 mmol/L), the contact angles on PTFE were larger than those on paraffin films, possibly due to the lower surface energy of PTFE (PTFE: $\gamma_{c}$ = 18 mN/m; paraffin films: $\gamma_{c}$ = 26 mN/m). According to Young’s equation $\gamma_{sg}$ − $\gamma_{sl}$ = $\gamma_{lg}cos\theta$ ($\gamma_{sg}$ is the solid/gas interfacial energy, $\gamma_{sl}$ is the solid/liquid interfacial energy, and $\gamma_{lg}$ is the gas/liquid interfacial energy), therefore, the contact angle of the PFPE-C aqueous solution on PTFE is larger. Contrarily, at higher concentrations of PFPE-C (such as, 2 mmol/L), the contact angles on PTFE were smaller than those on paraffin films, possibly due to the high structural similarity of the PFPE-C hydrophobic chain with PTFE. Surfactant molecules are more easily ranked on PTFE [32,33]. Moreover, aqueous solutions of PFPE-C at concentrations higher than 0.1 mmol/L can almost wet PTFE and paraffin membranes, indicating that PFPE-C has superior wetting properties on low surface energy materials compared with conventional hydrocarbon surfactants.

## 4. Conclusions

This paper describes the synthesis of a novel quaternary surfactant (PFPE-C) by a two-step reaction involving amidation and quaternization, followed by its structural characterization by FT-IR and NMR. Experimentation indicates that the lowest aqueous-solution surface tension of PFPE-C at 25 °C is 17.35 mN/m, and its critical micellar concentration is 0.024 mmol/L. The micellization of PFPE-C in aqueous solutions is a spontaneous exothermic process, and the CMC increases with increasing temperature, as indicated by conductivity and thermodynamic analyses. Additionally, the adsorption of PFPE-C at the air–water interface is confirmed to be a hybrid kinetic-controlled adsorption. The contact angles of PFPE-C aqueous solutions on PTFE and paraffin film indicate that wetting can be achieved at concentrations higher than 0.1 mmol/L, which means that PFPE-C also has good wetting properties on the surface of hydrophobic materials. DLS and Cryo-TEM results showed that PFPE-C can self-assemble into hollow vesicles in aqueous solution, and the vesicle structure can be applied to mimic biofilm, control drug delivery, serve as a template for the preparation of nanomaterials, and provide a suitable microenvironment for biochemical reactions. Ultimately, primary biodegradation experiments indicate that 19% PFPE-C can be degraded within a week.

## Data Availability

Not applicable.

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
