# Peer review of "Surface Activity, Wetting, and Aggregation of a Perfluoropolyether Quaternary Ammonium Salt Surfactant with a Hydroxyethyl Group"

_molecules, 2023, doi:10.3390/molecules28207151_

Round 1
Reviewer 1 Report
The manuscript titled “Surface activity, wetting, and aggregation of a perfluoropoly ether quaternary ammonium salt surfactant with a hydroxy ethyl group” highlighted the design and synthesis of quaternary surfactant containing a hydroxy ethyl group and its surface properties. The topic is interesting and the manuscript is well structured; however, some concerns need to be addressed before the publication of this work.
1. Add integration values in 1H-NMR spectra of the intermediate and PFPE-C (Figure 2).
2. In the 1H-NMR spectra of the intermediate, the peak appeared around 13 ppm and the authors assign this peak as a signal of CF3COOH. From where does this peak come?
3. The comparison of PFPE-C with DHDAC is not logical since both have different hydrophobic parts even not a similar number of carbons and arrangement of groups in hydrophobic part.
4. The authors claimed that the CMC values increased on increasing the temperature. Is it not dissimilar to literature? Kindly add references to support these findings.

Author Response
请参阅附件。

Reviewer 2 Report
Dou et al. embarked on synthesizing a novel perfluoropolyether quaternary ammonium salt surfactant. While the micellization, aggregation, air-water interfacial adsorption, and wettability have been characterized thoroughly and elucidated in a comprehensible manner, the introduction could benefit from more in-depth details. Here are specific recommendations for improvement:
1. It is vital to provide explanations for any abbreviations introduced. In this regard:
- In the abstract, kindly clarify the terms "????" and "CMC".
- In the introduction on line 37, the abbreviation "PFOS" needs explanation.
2. The introduction seems to lack the depth that readers might expect:
- First paragraph: It would be beneficial to commence by offering a broad overview of surfactants. A logical progression might be to discuss the categories of surfactants, particularly emphasizing traditional hydrocarbon and perfluorinated types. Only after laying this foundation should the drawbacks of hydrocarbon be elaborated upon, leading into the extensive applications of fluorocarbon surfactants.
- Second paragraph: The emphasis here should be primarily on perfluorinated surfactants. An in-depth discussion might encompass the definition of perfluorinated surfactants, a glimpse into the typical chemical structures, and concerns regarding non-degradable alkyl chains.
- Third paragraph: It would be beneficial for readers if the significance of Wang et al.'s work concerning the present study was highlighted. If, as mentioned in the second paragraph, the inclusion of oxygen atoms aims to address biodegradability, why then shift the focus to a quaternary ammonium surfactant? Shedding light on the advantages of this shift would be invaluable.
3. Post-synthesis, the purity of the synthesized product is a pivotal metric. Could you elaborate on the purity attained? Furthermore, is there a requirement for additional purification to rid the product of lingering intermediates?
4. While the FT-IR spectrum of PFPE-C is a welcome inclusion, it might be enriched by:
- Adding the background spectrum of FT-IR for reference.
- Incorporating arrows to demarcate and highlight specific peaks in Figure 1, enhancing clarity and reader comprehension.
5. A conspicuous oversight seems to be the missing title for Table 2. A concise yet descriptive title would augment the table's context within the manuscript.
Round 2
Reviewer 2 Report
This manuscript is in a good format and ready for publication.
Author Response
非常感谢您的评论!